# Rollout of ShangRing circumcision with active surveillance for adverse events and monitoring for uptake in Kenya

Elijah Odoyo-June[1]*, Nandi Owuor[2], Saida Kassim[3], Stephanie Davis[4], Kawango Agot[5], Kennedy Serrem[3], George Otieno[6], Quentin Awori[7], Jonas Hines[4], Carlos Toledo[4], Catey Laube[2], Christine Kisia[8], Appolonia Aoko[1], Vincent Ojiambo[9], Zebedee Mwandi[2], Ambrose Juma[3], Bartilol Kigen[3]

**1** Division of Global HIV & TB, U.S. Centers for Disease Control and Prevention, Kisumu, Kenya, **2** Jhpiego, Nairobi, Kenya, **3** MOH-NASCOP National STD/AIDS Control Program, Ministry of Health, Nairobi, Kenya, **4** Division of Global HIV & TB, U.S. Centers for Disease Control and Prevention, Atlanta, GA, United States of America, **5** Impact Research and Development Organization, Kisumu, Kenya, **6** University of Maryland Baltimore, Migori, Kenya, **7** Population Council/Engender Health, Nairobi, Kenya, **8** WHO Kenya Office, Nairobi, Kenya, **9** USAID-Kenya East Africa, Nairobi, Kenya

* Yed0@cdc.gov

**Data Availability Statement:** All relevant data are within the manuscript and its Supporting Information files.

## Abstract

### Introduction

Since 2011, Kenya has been evaluating ShangRing device for use in its voluntary medical male circumcision (VMMC) program according to World Health Organization (WHO) guidelines. Compared to conventional surgical circumcision, the ShangRing procedure is shorter, does not require suturing and gives better cosmetic outcomes. After a pilot evaluation of ShangRing in 2011, Kenya conducted an active surveillance for adverse events associated with its use from 2016–2018 to further assess its safety, uptake and to identify any operational bottlenecks to its widespread use based on data from a larger pool of procedures in routine health care settings.

### Methods

From December 2017 to August 2018, HIV-negative VMMC clients aged 13 years or older seeking VMMC at six sites across five counties in Kenya were offered ShangRing under injectable local anesthetic as an alternative to conventional surgical circumcision. Providers described both procedures to clients before letting them make a choice. Outcome measures recorded for clients who chose ShangRing included the proportions who were clinically eligible, had successful device placement, experienced adverse events (AEs), or failed to return for device removal. Clients failing to return for follow up were sought through phone calls, text messages or home visits to ensure removal and complete information on adverse events.

### Results

Out of 3,692 eligible clients 1,079 (29.2%) chose ShangRing; of these, 11 (1.0%) were excluded due to ongoing clinical conditions, 17 (1.6%) underwent conventional surgery due

**Funding:** This work has been supported by the President's Emergency Plan for AIDS Relief (PEPFAR) through the Centers for Disease Control and Prevention (CDC), under the terms of Award numbers GH001953, GH001952, GH009163, GH001957; GH001948; GH001946. The funders had no role in study design, data collection and analysis, decision to publish, or preparation of the manuscript.

**Competing interests:** The authors have declared that no competing interests exist.

to lack of appropriate device size at the time of the procedure, 97.3% (1051/1079) had ShangRing placement. Uptake of ShangRing varied from 11% to 97% across different sites. There was one severe AE, a failed ShangRing placement (0.1%) managed by conventional wound suturing, plus two moderate AEs (0.2%), post removal wound dehiscence and bleeding, that resolved without sequelae. The overall AE rate was 0.3%. All clients returned for device removal from fifth to eleventh day after placement.

## Conclusion

ShangRing circumcision is effective and safe in the Kenyan context but its uptake varies widely in different settings. It should be rolled out under programmatic implementation for eligible males to take advantage of its unique benefits and the freedom of choice beyond conventional surgical MMC. Public education on its availability and unique advantages is necessary to optimize its uptake and to actualize the benefit of its inclusion in VMMC programs.

## Introduction

### Background

Devices for medical male circumcision (MMC) have the potential to accelerate provision of voluntary medical male circumcision (VMMC) services by reducing the time it takes to perform the procedure, simplifying the procedure thereby allowing task shifting to lower cadre health care workers and increasing uptake of MMC among men who are averse to conventional surgery [1]. In 2011 the World Health Organization (WHO) established a program for prequalification of male circumcision (MC) devices to promote and facilitate access to safe, appropriate and affordable MC devices of good quality in an equitable manner [2]. This was followed in 2012 by publication of the WHO framework for clinical evaluation of devices for MC after prequalification [3], which outlines the criteria and assessments that countries should fulfill before endorsing any device for widespread use in national VMMC programs for HIV prevention. The necessary assessments fall into three broad phases namely, implementation pilot, active adverse events (AE) surveillance and passive AE surveillance [3].

In June 2015, WHO prequalified the ShangRing device (Wuhu SNNDA Medical Treatment Appliance Technology Co, Ltd, Wuhu City, China) for use in circumcision of adolescent and adult males ages 13 years and older after determining that it meets international standards of quality, safety and demonstrated efficacy [4]. The device consists of two concentric plastic rings, the inner of which has a silicon lined non bio reactive surface with a shallow groove on its outer surface against which the outer ring clamps to crush the foreskin when closed [4]. The outer ring consists of two halves that are hinged together at one end and on the other, a ratchet mechanism for tight closure such that there is rapid compression of the foreskin between the two rings and occlusion of blood flow to distal tissues when locked. The ShangRing compression force is sufficient to prevent slippage of tissue so that the foreskin can be excised at the time of device placement [4].

Based on the WHO prequalification of ShangRing and results of pilot studies in Kenya and other African counties, which demonstrated its safety, ease of use and good cosmetic outcomes [5–11], the Kenya national VMMC technical working group endorsed its rollout under an active AE surveillance protocol. The surveillance activity was conducted according to the

WHO framework for clinical evaluation of MC devices [3] and was consistent with similar previous initiatives, notably by Mavhu et al. [12]. The goal was to assess the feasibility of ShangRing device use in Kenya's VMMC program based on better understanding of the potential clinical and operational challenges or opportunities associated with its widespread use in routine clinical settings. The purpose was to guide the Ministry of Health's decision on whether to adopt ShangRing or not. The specific objectives were to (1) provide circumcision using ShangRing device as an alternative to other available methods of circumcision for males 13 years or older seeking VMMC services, (2) detect any new or rare AEs associated with ShangRing based on accumulation of a larger sample in addition to those in earlier pilot studies, (3) determine the proportion of men who choose circumcision through ShangRing device in routine VMMC service delivery settings when it is offered as an equally effective alternative to conventional surgical methods and (4) generate consumption data for different device sizes to guide forecasting of the proportions of different ShangRing device sizes to be procured to meet Kenya's VMMC program needs.

## Materials and methods

### Evaluation sites

Six health facilities offering routine VMMC services in five counties with support from PEP-FAR:-Jaramogi Oginga Odinga Teaching & Referral Hospital (JOOTRH), Bondo Sub County Hospital (SCRH), Got Agulu Dispensary, Loco Dispensary, Khunyangu Sub District Hospital, Mbita District Hospital were selected to implement the active adverse events surveillance (AAES) for ShangRing circumcision in Kenya. Key criteria for site selection included (1) location within a densely populated catchment area with well-established VMMC services so travel time between service point and most clients' residences was short, (2) health care workers competent in conventional surgical MC available to be trained on the ShangRing method, and (3) access to county or sub-county referral hospitals with skilled providers to manage rare complications. These facilities were selected in five counties:-(Siaya, Kisumu, Busia, Homabay, and Nairobi) to capture variations in VMMC service delivery contexts in Kenya. Each site was to contribute about 167 procedures towards an overall target of at least 1,000 ShangRing circumcisions.

### Training

A total of 21 certified VMMC surgeons were trained to perform circumcision using ShangRing. The training included a 2-day didactic session covering theory, AE management and documentation plus 1-day clinical observation of procedures performed by the trainers. Subsequently, each trainee performed 8 placements and 8 removals to attain certification.

Non-participating health care workers within the selected health facilities and from surrounding sites were sensitized on ShangRing AAES to enable them support and refer clients appropriately. Sensitization sessions were in form of half day meetings with the management and staff of all health facilities within the target area. Trained mobilizers sensitized the general public and potential VMMC clients on the availability of ShangRing as an alternative to conventional surgical circumcision.

### Recruitment

Healthy males aged 13 years or older seeking VMMC at the sites implementing ShangRing AAES or their outreach points were counseled and voluntarily tested for HIV and these testing negative were offered ShangRing as an alternative to conventional surgical MC. All ShangRing

circumcisions were performed at the designated fixed sites according to eligibility criteria and guidance in the product user manual [4]. Clients who chose ShangRing at outreach points were transported to the fixed site for the procedure.

Clients who chose ShangRing but were found clinically ineligible for the device (due to conditions like adhesions and thick or short foreskins) while remaining eligible for surgery were circumcised through conventional surgery according to the WHO and Kenya clinical manual for male circumcision under local anesthesia [13, 14].

## Procedure

Prior to device placement, clinically eligible clients who chose ShangRing gave consent for the procedure including permission for active follow up through telephone or home visits and for use of their records in the active AE surveillance. They underwent the procedure conducted by trained clinicians using the flip technique with injectable local anesthesia per the manufacturer's instructions for use [4]. Clients were discharged with instructions to keep the wound clean and dry, abstain from sex until complete wound healing, return on day seven for device removal and to contact the clinic or return at any time in case of concerns. Clients who failed to return for device removal on day seven or any other appointment were traced actively through phone calls and text messaging starting at close of business on the date of missed appointment until the end of day eight. Clients not successfully traced by the end of day eight would be traced physically through home visits and the outcomes documented. Similar to conventional surgical circumcision, AEs were classified based on the PSI/WHO Adverse Event Action Guide for VMMC [15] and moderate or severe AEs were reported then included in calculations of AE rate.

## Outcome measures

The following outcome measures were used to gauge safety, uptake and operational challenges that may be associated with widespread use of ShangRing:

1. AEs

   a. Rate of moderate or severe AEs, among clients circumcised through ShangRing including final outcomes in those experiencing the AEs

   b. Frequency and detailed characteristics of AEs not previously encountered with the device and final outcomes in those experiencing such AEs

2. Uptake

   a. Proportion of HIV uninfected males 13 years or older who chose ShangRing when offered as an alternative to conventional surgery. The numerator was all clients circumcised through ShangRing plus those screened out due to clinical ineligibility and those who missed the correct device size. The denominator used to compute uptake was the sum of HIV-negative clients aged 13 years or older circumcised conventionally or through ShangRing, screened out for clinical ineligibility and those who lacked appropriate device (presuming that all were given a choice between ShangRing and conventional surgery). Reasons for choosing or declining ShangRing were not collected because this evaluation was implemented in routine service delivery settings in conditions that were unfavorable for collection of additional elaborate information.

3. Clinical ineligibility

Proportion of those who chose ShangRing that were found to be clinically ineligible

4. Effectiveness

Proportion of eligible clients that had successful device placement with complete excision of foreskin.

5. Follow up rate

Proportion of ShangRing clients who returned for device removal (outcomes of active follow up efforts were recorded for clients who failed to return for their appointments)

Additional outcomes collected to inform program planning included duration of procedure; time until return for removal, classified as timely (one day before until one day after the scheduled day 7 removal), early (up till five days after placement), or late (day 9 and beyond); and device size.

Data analysis was primarily descriptive, using totals and disaggregation as appropriate by site, client age and device specific variable including size and days from placement to removal. The relationship between device size and age was analysed using generalized additive models with a line of best fit plus confidence bands generated then overplayed on a scatter plot.

This surveillance activity was reviewed and approved by the Kenya ministry of Health plus the US Centers for Disease Control and Prevention (CDC), Center for Global Health (CGH) human research protection procedures and determined to be non-research (CDC CGH HSR Tracking # D-14-2015; 2016–173).

## Results

From December 2016 to August 2018, a total of 3,692 HIV negative males aged 13 years and older seeking VMMC at six health facilities across five counties in Kenya were offered ShangRing. Of these, 1,079 (29.2%) chose ShangRing while 2,613 opted for surgical circumcision. Client recruitment, screening and enrollment cascade is presented in a Consolidated Standards of Reporting Trials (CONSORT) Fig 1.

Out of 1,079 clients who chose ShangRing, 11 (1.0%) were screened out due to clinical reasons (1 adhesion, 2 thick foreskins; 3 short foreskins; 2 hypospadias; 1 sickle-cell disease; 1 urethral discharge and 1 genital herpes). Five out of the 11 clients found ineligible for ShangRing were however eligible for dorsal slit method and were circumcised surgically. Ten out of the 11 clients who were clinically ineligible for ShangRing were 18 years or older. Seventeen (1.6%) of ShangRing eligible clients crossed over to conventional surgery due to lack of appropriate device size at the time of the procedure. Overall, 97.4% (1,051/1,079) of the clients who chose ShangRing underwent device placement.

Table 1 shows distribution by county, facility and age for the 1,051 clients who underwent ShangRing placement. The age range was 13–64 years with a median of 20 years (IQR 16–29).

ShangRing client recruitment, screening and enrollment cascade by county and site is presented in Table 2.

### Uptake of ShangRing

The overall proportion of eligible clients who chose ShangRing was estimated to be 29.2% but varied widely from 11.2% in Loco Dispensary, Nairobi to 97.1% at Bondo Sub County Hospital (Table 2).

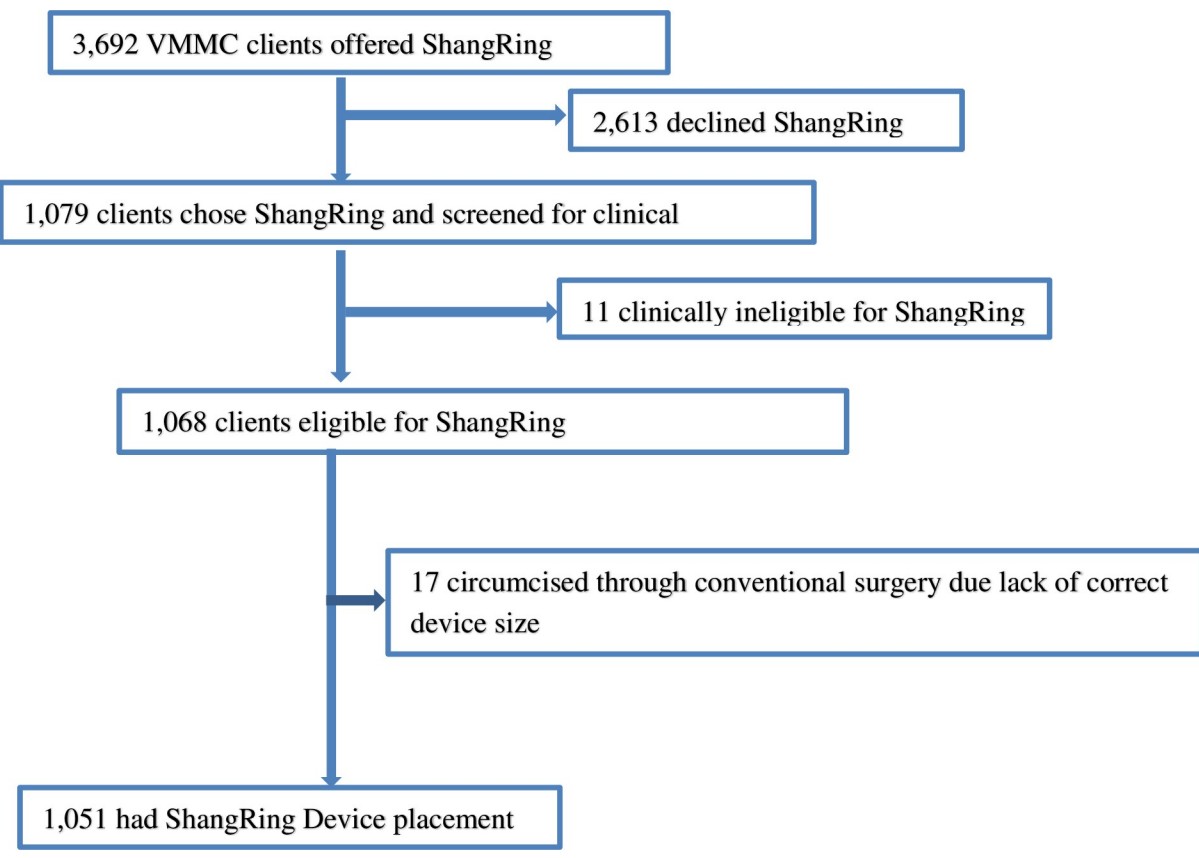

**Fig 1. Flow diagram for client screening and enrolment for ShangRing circumcision.**

### Adverse Events

Of the 1,051 clients who underwent ShangRing procedure, one had severe and two had moderate AEs for an overall AE rate of 0.3% (3/1,051). The first AE was a failed device placement due to incompletely locked outer ring noticed after excision of the foreskin (classified as a severe AE) managed by immediate removal of the device with completion of the procedure through conventional surgical suturing. The second AE, classified as moderate, was wound dehiscence after device removal and the third AE, also moderate, was immediate post-operative bleeding,

**Table 1. Distribution of clients circumcised through ShangRing by county, facility, and age band (N = 1,051).**

| County | Facility | Age bands in years | | | | | | | | | Total | Median age (IQR) |
|---|---|---|---|---|---|---|---|---|---|---|---|---|
| | | 13–14 | 15–19 | 20–24 | 25–29 | 30–34 | 35–39 | 40–44 | 45–49 | 50 | | |
| **Kisumu** | JOOTRH | 7 | 33 | 52 | 32 | 21 | 9 | 7 | 0 | 0 | 161 | 23 (20–29) |
| **Siaya** | Got Agulu Dispensary | 19 | 66 | 18 | 10 | 9 | 3 | 6 | 0 | 0 | 131 | 18(15–24) |
| | Bondo SCRH | 42 | 97 | 33 | 24 | 13 | 7 | 8 | 5 | 3 | 232 | 18 (15–24) |
| **Homabay** | Mbita District Hospital | 47 | 88 | 18 | 9 | 7 | 6 | 3 | 0 | 2 | 180 | 18 (14–19.8) |
| **Busia** | Khunyangu Sub District Hospital | 32 | 62 | 21 | 19 | 42 | 21 | 8 | 1 | 3 | 209 | 23 (15–32) |
| **Nairobi** | Loco Dispensary | 2 | 13 | 42 | 23 | 32 | 7 | 11 | 3 | 5 | 138 | 26 (21–33) |
| | **Total** | **149 (14%)** | **359 (34%)** | **184 (18%)** | **117 (11%)** | **124 (12%)** | **53 (5%)** | **43 (4%)** | **9(1%)** | **13 (1%)** | **1,051** | **20 (16–29)** |

**Table 2. ShangRing client recruitment, screening and enrollment cascade.**

| County | Kisumu | Siaya | | Homabay | Busia | Nairobi | All |
|---|---|---|---|---|---|---|---|
| Facility | JOOTRH | Got Agulu SCRH | Bondo SCRH | Mbita District Hospital | Khunyangu Sub District hospital | Loco Dispensary | Total |
| Clients offered SR | 678 | 157 | 242 | 1016 | 366 | 1239 | 3692 |
| Clients who chose SR n (%) | 163 (24.0%) | 133(84.7%) | 235(97.1%) | 193(19.0%) | 216(59.0%) | 139(11.2%) | 1079 (29.2%) |
| Accepted SR but clinically ineligible for SR | 1(0.6%) | 1(0.8%) | 2(0.9%) | 2(1.0%) | 4(1.9%) | 1(0.7%) | 11(1.0%) |
| Lacked correct device size | 1 | 1 | 1 | 11 | 3 | 0 | 17 |
| Circumcised through SR | 161 | 131 | 232 | 180 | 209 | 138 | 1051 |

SR = ShangRing; JOOTRH = Jaramogi Oginga Odinga Teaching and Referral Hospital; SCRH = Sub County Referral Hospital

managed through sustained application of pressure with the device in place. All AEs resolved without long term sequelae. There were no previously undescribed ShangRing related AEs.

## Average duration of device placement

Mean duration of device placement (measured from placement of the inner ring to complete excision of the foreskin) was 9.8 minutes (SD 2.8) and did not vary substantially by facility, client age or provider experience (not shown).

## Follow up rate among clients circumcised through ShangRing device

Of the 1,051 clients enrolled, one had immediate device removal as described above. All other 1,050 clients returned for device removal from the 5th to 11th post-placement day. There were two (0.2%) early removals before day six, 1,033 (98.3%) timely removals from the sixth to eigth day and 16 (1.5%) late removals from day nine to day 11 (Fig 2). All 43 clients who failed to return for removal by the 7th post-placement day were successfully reached by phone and returned for device removal within 4 days without physical tracing.

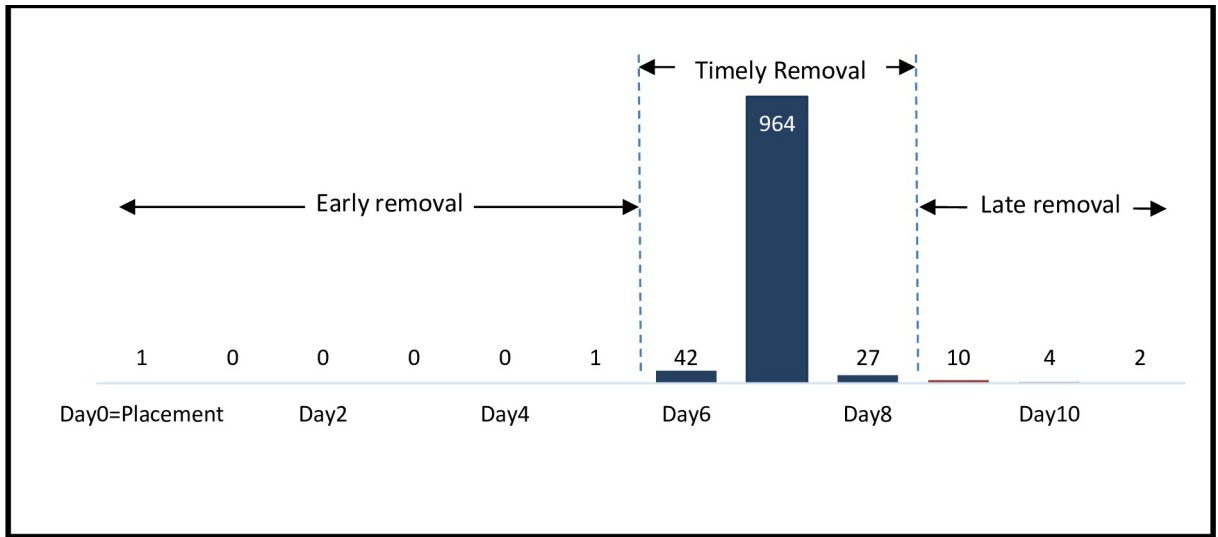

**Fig 2. Number of ShangRing removals by day since replacement.**

## Distribution of ShangRing device sizes used by client ages

The complete range of 18 available ShangRing device sizes for adolescents and adults, ranging from S (18mm) to A (40mm) were used in this evaluation (See Table 3). Five out of the six AAES facilities experienced at least one instance of stock out of some frequently used device

**Table 3. ShangRing device sizes used among Kenyan males 13-64yrs (n = 1051).**

| Device size in mm | Age bands in years | | | | | | | | | Row Totals |
|---|---|---|---|---|---|---|---|---|---|---|
| | 13–14 | 15–19 | 20–24 | 25–29 | 30–34 | 35–39 | 40–44 | 45–49 | 50+ | |
| A-40 | 2 | 15 | 18 | 13 | 17 | 4 | 3 | 0 | 0 | 72 |
| | 1.3% | 4.2% | 9.8% | 11.1% | 13.7% | 7.3% | 7.0% | 0.0% | 0.0% | 6.8% |
| A1-39 | 0 | 2 | 9 | 1 | 3 | 0 | 1 | 0 | 0 | 16 |
| | 0.0% | 0.6% | 4.9% | 0.9% | 2.4% | 0.0% | 2.3% | 0.0% | 0.0% | 1.5% |
| A2-38 | 0 | 0 | 1 | 1 | 1 | 0 | 0 | 1 | 0 | 4 |
| | 0.0% | 0.0% | 0.5% | 0.9% | 0.8% | 0.0% | 0.0% | 11.1% | 0.0% | 0.4% |
| A3-37 | 0 | 0 | 1 | 2 | 5 | 1 | 1 | 0 | 1 | 11 |
| | 0.0% | 0.0% | 0.5% | 1.7% | 4.0% | 1.8% | 2.3% | 0.0% | 8.3% | 1.1% |
| A4-36 | 0 | 2 | 2 | 1 | 3 | 0 | 0 | 0 | 0 | 8 |
| | 0.0% | 0.6% | 1.1% | 0.9% | 2.4% | 0.0% | 0.0% | 0.0% | 0.0% | 0.8% |
| B-35 | 1 | 19 | 17 | 10 | 15 | 8 | 2 | 1 | 0 | 73 |
| | 0.7% | 5.3% | 9.2% | 8.6% | 12.1% | 14.6% | 4.7% | 11.1% | 0.0% | 6.9% |
| C-34 | 2 | 29 | 23 | 12 | 20 | 8 | 2 | 0 | 0 | 96 |
| | 1.3% | 8.1% | 12.5% | 10.3% | 16.1% | 14.6% | 4.7% | 0.0% | 0.0% | 9.1% |
| D-33 | 6 | 44 | 21 | 21 | 6 | 14 | 10 | 0 | 2 | 124 |
| | 4.0% | 12.3% | 11.4% | 18.0% | 4.8% | 25.5% | 23.3% | 0.0% | 16.7% | 11.8% |
| E-32 | 15 | 51 | 24 | 18 | 13 | 8 | 7 | 2 | 6 | 144 |
| | 10.07% | 14.21% | 13.04% | 15.38% | 10.48% | 16.36% | 16.28% | 22.22% | 50.00% | 13.78% |
| F-31 | 9 | 33 | 15 | 6 | 13 | 4 | 6 | 2 | 1 | 89 |
| | 6.0% | 9.2% | 8.2% | 5.1% | 10.5% | 7.3% | 14.0% | 22.2% | 8.3% | 8.5% |
| G-30 | 12 | 28 | 22 | 11 | 12 | 3 | 8 | 0 | 2 | 98 |
| | 8.1% | 7.8% | 12.0% | 9.4% | 9.7% | 5.5% | 18.6% | 0.0% | 16.7% | 9.3% |
| H-29 | 13 | 34 | 14 | 8 | 7 | 2 | 1 | 1 | 0 | 80 |
| | 8.7% | 9.5% | 7.6% | 6.8% | 5.7% | 3.6% | 2.3% | 11.1% | 0.0% | 7.6% |
| I-28 | 24 | 51 | 6 | 12 | 7 | 2 | 0 | 2 | 0 | 104 |
| | 16.1% | 14.2% | 3.3% | 10.3% | 5.7% | 3.6% | 0.0% | 22.2% | 0.0% | 9.9% |
| K-26 | 14 | 23 | 7 | 1 | 2 | 0 | 2 | 0 | 0 | 49 |
| | 9.4% | 6.4% | 3.8% | 0.9% | 1.6% | 0.0% | 4.7% | 0.0% | 0.0% | 4.7% |
| M-24 | 4 | 8 | 3 | 0 | 0 | 0 | 0 | 0 | 0 | 15 |
| | 2.7% | 2.2% | 1.6% | 0.0% | 0.0% | 0.0% | 0.0% | 0.0% | 0.0% | 1.4% |
| O-22 | 18 | 9 | 0 | 0 | 0 | 0 | 0 | 0 | 0 | 27 |
| | 12.1% | 2.5% | 0.0% | 0.0% | 0.0% | 0.0% | 0.0% | 0.0% | 0.0% | 2.6% |
| Q-20 | 13 | 5 | 1 | 0 | 0 | 0 | 0 | 0 | 0 | 19 |
| | 8.7% | 1.4% | 0.5% | 0.0% | 0.0% | 0.0% | 0.0% | 0.0% | 0.0% | 1.8% |
| S-18 | 16 | 6 | 0 | 0 | 0 | 0 | 0 | 0 | 0 | 22 |
| | 10.7% | 1.7% | 0.0% | 0.0% | 0.0% | 0.0% | 0.0% | 0.0% | 0.0% | 2.1% |
| COLUMN TOTALS | 149 | 359 | 184 | 117 | 124 | 54 | 43 | 9 | 12 | 1,051 |
| | 14.2% | 34.1% | 17.5% | 11.1% | 11.8% | 5.2% | 4.1% | 0.9% | 1.1% | 100.0% |

The device size used increased with advancing age from 13–20 years ($\chi^2$ = 0.47, p<0.001) then plateaued from 20–40 years ($\chi^2$ = 0.057, p = 0.224) and above 40 years ($\chi^2$ = 0.006 p = 0.9686). Beyond 45 years, the relationship between device size and age may have been obscured due to the small number of clients in this age bracket.

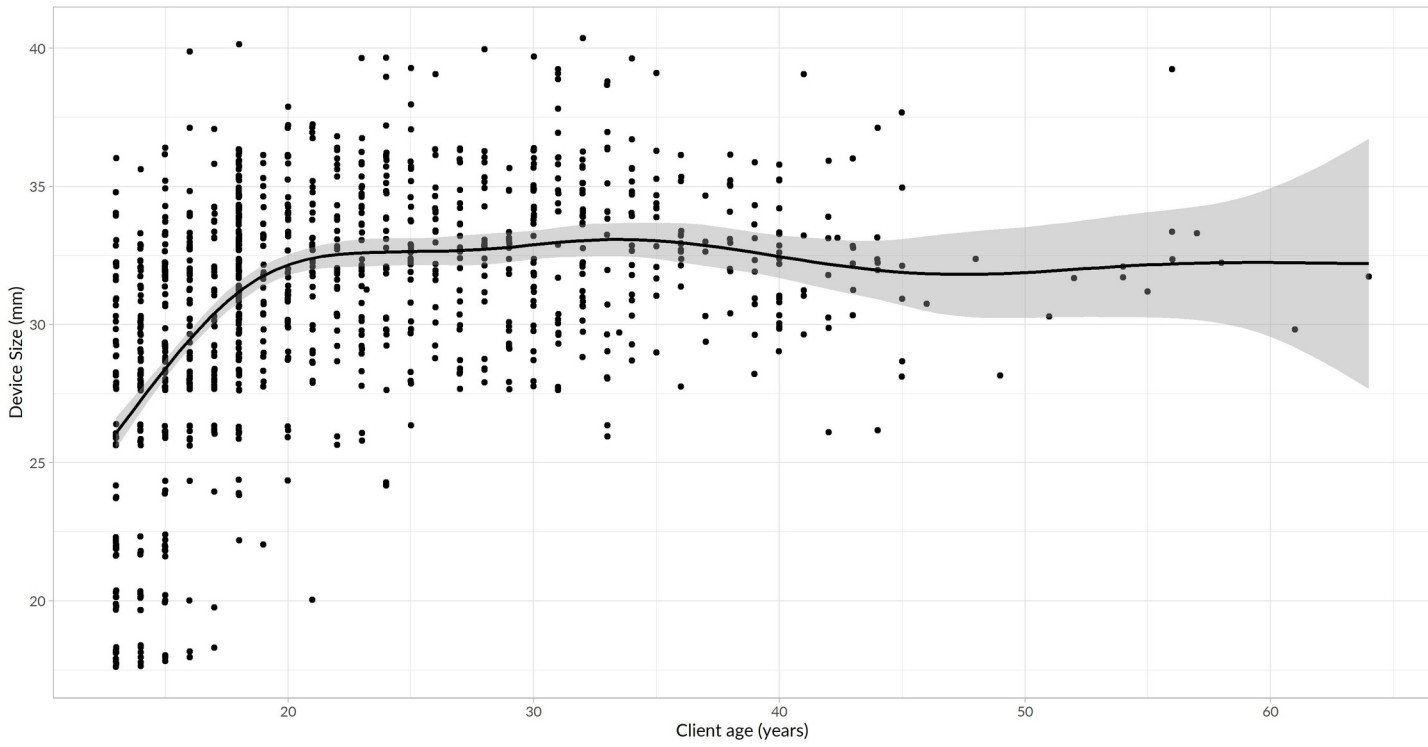

**Fig 3. Distribution of ShangRing device sizes used by client age.**

sizes especially D to G which collectively accounted for 43.% of the total devices used. Overall, 17 eligible clients who chose ShangRing crossed over to conventional surgical circumcision due to stock out of the appropriate device size.

A graphic presentation of the relationship between device size and client age is presented in Fig 3.

## Discussion and conclusion

### Discussion

This evaluation examined the safety, uptake and overall feasibility of ShangRing device for use in Kenya's VMMC program based on 1,051 procedures conducted in routine health care settings. Overall, the device was effective, safe, with limited operational bottlenecks for roll out but had wide variation in uptake across different sites. The results corroborate findings from previous studies in Uganda, Kenya and Zambia [8, 9, 10] which showed that the device is effective, safe and client compliance with appointments for device removal on day seven is excellent.

The percentage of clients for whom the device is not suitable was quantified. Only 11 (1%) of the 1,079 clients who chose ShangRing were found to be clinically ineligible for the device and of these, five were moreover eligible for surgery leaving only five clients ineligible for both ShangRing and conventional surgery. Comparable low rates of clinical ineligibility for the device have also been reported in other studies; two out of 500 (0.4%) in Zambia [16], one out 200 (0.25) in Kenya [13] and five out of 1,211 (0.4%) men in a field study in Kenya and Zambia [9]. This is reassuring because only a small proportion of men will be clinically unsuitable for ShangRing circumcision as the device is rolled out, but it highlights the necessity of maintaining some availability of conventional surgical services.

The moderate and severe AE rate in this evaluation was low (0.3%) and corroborates reports from earlier studies across Africa [8, 9, 14]; suggesting that widespread use of ShangRing in the VMMC program would be safe.

The ShangRing uptake rate of 29.2% observed in this evaluation is much lower than was reported in Uganda by Kigozi et al. from comparable evaluations among adults (81.8%) and adolescents (82.8%) [8, 17]. Some previous studies reported high ShangRing acceptability rates based on client satisfaction with the procedure and whether they said they would recommend the procedure to friends [6, 11] which is not comparable to this evaluation. Possible explanations for the overall low uptake and its wide variation across sites (11.1%-97.2%) include variable recruitment practices across different counties with some sites only offering ShangRing at static facilities where both conventional surgery and ShangRing procedure would be performed on-site while others additionally offered ShangRing at outreach intake points from where clients would be transported to different fixed sites for the procedure. The need for transfer of ShangRing clients to a different location for placement may have discouraged clients served at outreach sites from choosing the device. The presentation of ShangRing as a different but equally effective alternative to conventional surgery rather than a superior option may not have been enough motivation for clients to switch from the widely known surgical methods to a new device. When people are offered something new and are expected to make a choice immediately, the natural tendency is to err on the side of caution. Finally, the computation of uptake is based on an assumption that all HIV negative clients aged 13 years or older who were circumcised through conventional surgery at the evaluation sites actively declined ShangRing without recording each individual's choice; it is possible that in high demand seasons, some clients slipped through to conventional surgery without being offered ShangRing leading to underestimation of uptake.

Eventually, uptake of ShangRing will likely increase through information diffusion as more men experience the clinical advantages of the device such as short procedure time, absence of suturing and better cosmetic outcomes. Further, implementation of the March 2019 WHO amendment of ShangRing prequalification to include use of topical anesthesia and no-flip technique [18] may stimulate incremental demand from men who dislike injection and prefer a faster procedure.

The complete range of 18 ShangRing device sizes for adolescents and adults ranging from S (18mm) to A (40mm) were used in this evaluation with 17 (1.6%) clients missing the correct device at the time of the procedure. This was the most significant logistical challenge experienced during this AAES. The need to continuously stock all adult and adolescent device sizes may pose a challenge for procurement because the relative consumption rate of different devices sizes depends on daily service uptake by different age bands which is unpredictable. Similar challenges of device stock outs have been reported in other ShangRing studies [12, 14] and may require large stock of buffer supplies in the roll out phase. An important new development that addresses this challenge is the approach of using reduced number of ShangRing sizes which was evaluated by Feldblum et al. in Zambia and found to be effective and safe [16]. The results showed that using half the number (every other size) of adult ShangRing device sizes is sufficient for safe service delivery. WHO subsequently amended the prequalification of ShangRing in March 2019 to include availability and use of every other device size; this may alleviate some of the challenges around need to stock a large number of different devices sizes [18].

## Limitations

ShangRing uptake in this evaluation should be interpreted with caution because it is based on clients who presented for VMMC at the health facilities implementing AAES and their outreaches; uptake by the unreached broader target communities may be different.

The observed wide variation in uptake of ShangRing across different sites cannot be explained fully because the reasons for not choosing ShangRing were not recorded. Additional efforts combining qualitative and mixed methods are therefore recommended to explore possible reasons for low and variable uptake of ShangRing. This will guide effective demand creation for the device in areas where uptake is low.

Overall, a trajectory of increasing uptake for ShangRing is likely because its prequalification has been updated to include no flip technique and topical anesthesia, which may attract more clients.

Another limitation is the documentation of AE rate based on client-provider interaction during placement and at day 7 removal only. The protocol did not include additional visits beyond day 7 removal, therefore any AEs occurring post-removal may have been missed if the client failed to seek help at the designated ShangRing evaluation facilities. This could result in underestimation of AE rate.

## Conclusion

ShangRing circumcision is effective and safe in the Kenyan context and should be rolled out under programmatic implementation for men to take advantage of its unique benefits and the freedom of choice beyond conventional surgical MMC. Public education on its availability and advantages is necessary to increase its uptake and realization of public health benefits of its inclusion in VMMC programs. The WHO amendment of its prequalification in March 2019 to include its use among younger adolescents 10–12 years, application under topical anesthesia and no-flip technique may stimulate incremental demand for VMMC.

## Disclaimer

The findings and conclusions in this paper are those of the authors and do not necessarily represent the official position of the funding agencies.

## Supporting information

**S1 Table. Dataset from ShangRing AAES in Kenya Line.**
(XLSX)

## Acknowledgments

We gratefully acknowledge the clients who participated in the active AE surveillance for ShangRing circumcision and the Kenya national Ministry of Health including the directors of health for Kisumu, Siaya, Homabay, Migori, Busia and Nairobi counties for providing oversight during the implementation of the active AE surveillance activity.

The authors appreciate male circumcision providers and various VMMC implementing partners including Impact Research and Development Organization (IRDO), ICAP at Colombia University, Eastern Deanery AIDS Relief Program (EDARP) and Center for Health Solutions (CHS) for supporting actual implementation in various facilities and for ensuring timely reporting of data. The authors are grateful to the VMMC service provision team members including Godfrey Onchiri (CHS-Kenya), Tuma Noah (CHS-Kenya), Rachel Odhiambo

(ICAP), Mourine Ongoro (ICAP), Charles Nderitu (EDARP-Kenya), Maurice Magudha (CMMB), Felix Njue (CMMB), Lilian Oduri(CMMB) among many others. Without their collective support, the active surveillance could not have been done.

Special thanks go to Jhpiego for providing technical and logistical support for the national coordination of the active AE surveillance and to the President's Emergency Plan for AIDS Relief (PEPFAR) for funding the active AE surveillance through Centers for Disease Control and Prevention (CDC). Finally we gratefully acknowledge Rinee Ridzon and Paul Musingila both of CDC Kenya for their review and assistance with editing the manuscript.

## Author Contributions

**Conceptualization:** Elijah Odoyo-June, Nandi Owuor, Kennedy Serrem, Quentin Awori.

**Data curation:** Nandi Owuor, Saida Kassim, George Otieno.

**Formal analysis:** Elijah Odoyo-June, Nandi Owuor, Saida Kassim, George Otieno.

**Investigation:** Nandi Owuor, Kennedy Serrem.

**Methodology:** Elijah Odoyo-June, Nandi Owuor.

**Supervision:** Elijah Odoyo-June, Nandi Owuor, Kennedy Serrem.

**Validation:** Elijah Odoyo-June, Nandi Owuor.

**Writing – original draft:** Elijah Odoyo-June.

**Writing – review & editing:** Elijah Odoyo-June, Nandi Owuor, Stephanie Davis, Kawango Agot, Quentin Awori, Jonas Hines, Carlos Toledo, Catey Laube, Christine Kisia, Appolonia Aoko, Vincent Ojiambo, Zebedee Mwandi, Ambrose Juma, Bartilol Kigen.

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
