## [Decision Letter · Decision Letter 0]

9 Jul 2019

PONE-D-19-15563

Rollout of ShangRing circumcision with active surveillance for adverse events and monitoring for acceptability in Kenya

PLOS ONE

Dear Dr Odoyo-June,

Thank you for submitting your manuscript to PLOS ONE. After careful consideration, we feel that it has merit but does not fully meet PLOS ONE’s publication criteria as it currently stands. Therefore, we invite you to submit a revised version of the manuscript that addresses the points raised during the review process.

We would appreciate receiving your revised manuscript by Aug 23 2019 11:59PM. To enhance the reproducibility of your results, we recommend that if applicable you deposit your laboratory protocols in protocols.io, where a protocol can be assigned its own identifier (DOI) such that it can be cited independently in the future. For instructions see: http://journals.plos.org/plosone/s/submission-guidelines#loc-laboratory-protocols

We look forward to receiving your revised manuscript.

Kind regards,

Webster Mavhu

Academic Editor

PLOS ONE

Journal Requirements:

2.

We suggest you thoroughly copyedit your manuscript for language usage, spelling, and grammar. If you do not know anyone who can help you do this, you may wish to consider employing a professional scientific editing service.  

3. In your Methods section and Ethics Statement, please specify whether data were obtained in a fully anonymized and de-identified manner, and whether any of the researchers/authors had access to identifying information. If any of the authors had access to identifying information (names, addresses, etc.), please explain whether you obtained participant consent, or whether the requirement for informed consent was waived.

4. Thank you for including your ethics statement: "This surveillance activity was reviewed by the the Kenya ministry of Health VMMC program plus the Centers for Disease Control and Prevention (CDC), Center for Global Health (CGH) human research protection procedures and determined to be nonresearch (CDC CGH HSR Tracking # D-14-2015; 2016-173)."

a) Please amend your current ethics statement to confirm that your named institutional review board or ethics committee specifically approved this study.

Additional Editor Comments (if provided):

This is a well-written manuscript describing active surveillance of the ShangRing device as per the WHO framework. I have a few comments:

1) In response to reviewers, authors should state that conduct of the active surveillance was consistent with the WHO framework and similar, previous initiatives (e.g. Mavhu 2016; 2019). The working definition of "acceptability" is consistent with both WHO and similar, previous initiatives. Of note active surveillance is not a study per se. Other reviewers comments are however, relevant and need to be addressed.

2) Possible reasons for variability in acceptability (11-97%) need to be provided. In other instances, this was because providers were actively sabotaging a device as more VMMCs in a shorter time meant less remuneration. If possible reasons are unavailable, this should be stated as a limitation and future initiatives should explore these. If it was due to absence if systematic demand creation, a recommendation would be that device-specific demand creation should be intensified and/or target those sites/communities with low acceptance.

3) Also, future initiatives (including qualitative/mixed methods) should explore possible reasons for not taking up ShangRing - these will be critical in informing demand-creation for the device.

4) Active tracing procedures need to be described in detail (see previous device active surveillance papers) - How many call/text attempts and when?

5) Rest of comments are editorial - most have been raised by reviewers

.Authors vacillate between UK & US English (e.g. anesthesia/anaesthesia, program/programme, analysed) - should use one consistently

.References - 101, 104, 304, 333-334

.Line 71 - 12 years

.112-studies, (13)

.167 - of ShangRing?

.196 timely, (comma use)

.276 - is may?

.330 AES or AAES?

.347, 361 (10-12 years)

Reviewers' comments:

Reviewer's Responses to Questions

**Comments to the Author**

1. Is the manuscript technically sound, and do the data support the conclusions?

Reviewer #1: Partly

Reviewer #2: Partly

2. Has the statistical analysis been performed appropriately and rigorously? 

Reviewer #1: I Don't Know

Reviewer #2: Yes

3. Have the authors made all data underlying the findings in their manuscript fully available?

Reviewer #1: Yes

Reviewer #2: Yes

4. Is the manuscript presented in an intelligible fashion and written in standard English?

Reviewer #1: Yes

Reviewer #2: Yes

5. Review Comments to the Author

Reviewer #1: Major comments:

This is an interesting paper on safety and acceptability of ShangRing in Kenya. It is clear that devices are an important part of VMMC scale up and efficiency. The paper would be of interest to your readers and is timely. The data could inform a critical component of HIV prevention programs.

However with the multiple purposes of this evaluation including assessment of ShangRing safety and acceptability, the authors fail to completely address that they could not demonstrably show acceptability. Safety was assessed and ShangRing demonstrated safe. An average acceptability of only 29% is poor – half of the 6 sites had acceptance rates of 25% or less. This result, in contrast to previous research, is diminished and ShangRing scale up repeatedly suggested, although premature based on these data. The abstract, discussion, and conclusions are, therefore, misleading. As acceptability was not demonstrated, the investment in training, devices, and alternatives to surgical MC for ShangRing roll out are not supported in this paper. Related, the results – only one sentence on acceptability (a major outcome) are not well balanced. It is explained in the methods and limitations that they did not collect information from clients on reasons not to want ShangRing as this was largely routine data collection; however, this major limitation should be better explained. Although they posit some potential explanations in a discussion paragraph, the lack of further exploration of this interesting result severely limits the utility of this research.

Attention and revision with an eye to the results on acceptability is needed to strengthen the paper. The authors should return to clinics, focal persons, or other key VMMC informants to illuminate these findings and give some further rational for this interesting result. This is the key piece of the paper that could inform scale in Kenya and the region. This would not take long and would help make the results more informative.

Minor comments:

Line 101: Fix references. This is distracting and poor editing: “other African counties, which demonstrated its safety, ease of use and good cosmetic outcomes [5, 6, 7, 8, 9 102 10, 11, 12], the Kenya national VMMC technical working group endorsed 103 its rollout under an active AE surveillance protocol in line with the WHO framework for clinical 104 evaluation of MC devices[Error! Bookmark not defined.].” Line 304/334 reference as well.

Line 114: Rephrase: It is not an “equivalent alternative.” This phrase seems misleading.

Line 134: Please provide more details on the training ShangRing. How long was training? How was proficiency attained? How many procedures did the trainees watch or perform? Was AE identification, treatment, management, and documentation included?

Please clarify. Evaluation sites were noted to be chosen for dense populations with persons living close to MC sites. However, you later note in lines 155+, that clients receiving ShangRing were transported from additional outreach intake points to the fixed sites. This seems to contradict your evaluation site criteria.

Line 161: This is the basis for the paper and needs more detail in this section. How was active surveillance conducted and when? Were phone calls the first line of active surveillance and then home visits? Or either? Or both? Were they triggered on Day 7 or Day 8? Please clarify the active tracing procedures.

Line 190: Follow up rate outcome includes outcomes of active follow up efforts for clients who failed to return for follow up. Does this include only those with observed outcomes or also those who reported outcomes by phone?

211: This is a large limitation to the usefulness of the study to inform scale up. “Reasons for choosing or declining ShangRing were not collected because this evaluation was implemented in routine service delivery settings.”

Table 2: Why do you think that Mbita District Hospital had the majority of the missing ShangRIng sizes?

238: The fact that acceptability of ShangRing varied widely is suggestive of highly varied recruitment or demand creation procedures. How were demand creation officers trained? How did that training vary across sites? Were the mobilizers paid to recruit? Was that the same across sites?

259: No need to repeat outcome measures of timing.

Line 267 spelling error, “avialble” is an example of the need for grammar, spacing, and spelling review throughout. There are many examples of sloppy copy editing. Another: Line 276, “and age is may have been obscured due to the small number of clients in this age bracket.” Line 330 “experienced during this AES.” Even ShangRing is not consistently spelled throughout (check your tables and figures). Many more missing punctuation and errors to be addressed that show lack of close editing review.

Results lines 300+. This lower acceptability seems like the major result. This paragraph could be strengthened. What different in lower and higher acceptance sites? Just urban/rural? Was it only lack of ShangRing availability at the sites? If the transportation was a factor, this also shows that the sites were poorly selected as the criteria for inclusion suggested that close proximity to sites was a reason for site selection. Please explain.

Reviewer #2: Thank you to the authors for the manuscript that focuses on the active surveillance of the Shang Ring device in Kenya across 6 sites in 5 counties. While I consider the manuscript to be valuable it does require some revision and points of clarity.

Abstract

1. Conclusion in the abstract reads as a direct copy of the main manuscript conclusion.

Background

1. Where it is written medical circumcision recommend amending to read medical male circumcision (MMC) to be specific

2. There are sentences that require references and reference correction: end of sentence line 88, end of sentence line 92, end of sentence line 94, end of sentence line 97, reference 9 in line 102, reference error in line 104.

3. The authors mention that the goal of the surveillance activity was to “assess the feasibility of Shang Ring…” however, as per the WHO guidelines on evaluating medical male circumcision devices, feasibility is usually assessed during the pilot stage. Therefore, was this a true feasibility assessment? Furthermore, it is not clear what the operational challenges or opportunities were within the broader scope of the manuscript.

4. Were the outlined objectives representative of the overall active surveillance or the manuscript? Recommend revising these objectives to outline the aims of the manuscript under background and the active surveillance objectives can be included under the methods section. However, as it stands the presented objectives do not speak to the results of the manuscript. An example: objective (2) “detect new or rare AEs…” nothing new or rare was reported. None of the objectives clearly talk to operational challenges or operational bottlenecks. Based on the discussion, one can make the assumption that objective 4 makes reference to operational challenges however the link is not clear.

Methods

1. Were the health facilities or counties supported by PEPFAR. Consider re-wording for clarity.

2. Selection criteria for sites: the authors list 3 however it reads as 4. Population density and travel time should be two separate criteria.

3. What was considered to be a ‘short’ travel time between site and residence?

4. How did the authors define a ‘competent’ healthcare worker in conventional surgical MMC?

5. The target was 167 per site, however some were under and others over? Was the target enrolment controlled across sites? Did all sites start initiation of the Shang Ring at the same time?

6. Under training, the authors touch on sensitization of non-participating health care workers and the general public how was this achieved, through workshops, information sessions, pamphlets…

7. The study only enrolled HIV uninfected clients, does this mean that clients were tested at the facility prior to being offered the Shang Ring or was HIV status considered under screening?

8. Line 146 to 149 is unclear, recommend rephrasing to: Clients who chose ShangRing but were found clinically ineligible for the device (due to conditions like adhesions and thick or short foreskins) while eligible for surgery were circumcised through conventional surgery according to the Kenya clinical manual for male circumcision under local anaesthesia (13).

9. Under procedure it mentions that clients consent to “active follow-up” however the study does not continue to follow clients after device removal. This needs to be clarified.

10. Provide additional information to explain why some clients had to be transported to a service delivery point to receive the Shang Ring. Was this the same at those sites for clients opting for conventional MMC?

11. How was wound healing determined?

12. There is no description of how the data was collected and captured. Were these clinic files that were reviewed daily, logs that were assessed retrospectively and where was the information captured and stored?

Outcome measures

13.1. Proportion of AEs was calculated by adding the severe and moderate not severe or moderate

13.2. Under outcomes: safety is reported by the AE rate, acceptability is reported by device uptake. However, it is not clear how clinical eligibility, effectiveness, follow-up rate, device placement duration are related to operational challenges.

13.3. Does time until return for removal fall under follow up rate? This is confusing considering that the study has no formal follow-ups

14. Data analysis section should be a separate paragraph

15. Provider experience as a measurement only features under analysis description and does not appear in the results, discussion or background. Not clear how it is relevant.

Results

1. The study dates in the abstract are different from the dates in the results

2. In line 211-212 authors provide justification for why acceptability data is not collected, this is misplaced and should be moved from results to methods.

3. Figure 1: 11 clinically ineligible with not description in figure for why however 17 circumcised through conventional surgery with a description for why. Be consistent with how you represent your information.

4. What happened to the other 6 who were ineligible for Shang Ring and conventional surgical circumcision, were they referred for care?

5. Move sentence “Ten out of the 11 clients who were…” (line 222-223) before sentence starting with “Seventeen (1.6%)…” (line 220)

6. Be consistent with decimals when reporting percentages 1.6% vs 97%

7. It is not clear what the authors mean by “There were no previously undescribed ShangRing related AEs” (line 250-251)

8. What is the significance on reporting on duration of device placement? Device placement is not included in the discussion section. Furthermore, there is no data provided on duration of device removal.

9. Check Table of ShangRing device sizes: in table, A4 is 36 mm while in text A4 is 40mm.

10. Figure 3: The distribution of device sizes by age using the scatter plot is good. As the authors describe the plot is biphasic (13-20 years) and 20+ years regarding distribution of need for different device sizes. Device size variability is required in the younger age categories.

Discussion

1. Consider re-working the sections of the discussion to follow the order in which the results are reported. Current order of results: Eligibility, Acceptability, AE, Placement duration, follow-up rate, size distribution. The discussion sections are: Eligibility, AE, Acceptability, Size.

2. The authors spend a considerable length of the discussion justifying the reasons for not collecting additional information on acceptability for this reason there are certain sections of the discussion that read like they should be under the ‘Limitations’ heading. A recommendation would be to change the wording from acceptability and instead report on uptake of the ShangRing device with a future recommendation for more detailed acceptability studies to be done.

3. The discussion section focuses too much on the 17 cases that did not have the appropriate device sizes available. However, there is no description in the results of what these sizes were or the age categories of the clients. The manner in which the device size is represented in the discussion section renders the scatter plot irrelevant. Recommend finding a way to report on the sizes that were relevant to the active surveillance.

4. The discussion section would benefit from additional literature to substantiate some of the points being made by the authors.

Limitations

1. The description of no additional visits should also be reflected in the methodology

Conclusion

1. Remove references from conclusion

2. The WHO amendment to the prequalification is mentioned in the Limitations and Conclusions section however it is not in the Discussion section. It’s link to the overall paper is not clear.

Acknowledgement

1. Acknowledge the clients who participated in the active surveillance

6. PLOS authors have the option to publish the peer review history of their article (what does this mean?). If published, this will include your full peer review and any attached files.

Reviewer #1: No

Reviewer #2: No

---

## [Author Response · Author response to Decision Letter 0]

3 Sep 2019

Point by point response to the editor and the two reviewers are presented in a single table divided into three sections each addressing a single reviewer. The complete response is uploaded as Response to Reviewers.

---

## [Editor Report · Decision Letter 1]

9 Sep 2019

PONE-D-19-15563R1

Rollout of ShangRing circumcision with active surveillance for adverse events and monitoring for uptake in Kenya

PLOS ONE

Dear Dr Odoyo-June,

Thank you for submitting your manuscript to PLOS ONE. After careful consideration, we feel that it has merit but does not fully meet PLOS ONE’s publication criteria as it currently stands. Therefore, we invite you to submit a revised version of the manuscript that addresses the points raised during the review process.

We would appreciate receiving your revised manuscript by Oct 24 2019 11:59PM. To enhance the reproducibility of your results, we recommend that if applicable you deposit your laboratory protocols in protocols.io, where a protocol can be assigned its own identifier (DOI) such that it can be cited independently in the future. For instructions see: http://journals.plos.org/plosone/s/submission-guidelines#loc-laboratory-protocols

We look forward to receiving your revised manuscript.

Kind regards,

Webster Mavhu

Academic Editor

PLOS ONE

Additional Editor Comments (if provided):

All reviewer comments have been adequately addressed. A few editorial comments:

Line 139- Subsequently,

182 - ...who chose...

260 - ...third AE, also moderate, was...

270 Removals were... beyond). repeats lines 206-209 - could be paraphrased or left out

326 ...locations,...??

366 ...uptake of ShangRing...

382 ....realization of public...

384 ...adolescents 10-12 years...

---

## [Author Response · Author response to Decision Letter 1]

9 Sep 2019

Pint-by-point Response to Reviewer Comments

Protocol title: Rollout of ShangRing circumcision with active surveillance for adverse events and monitoring for uptake in Kenya. PONE-D-19-15563R1

Response to additional Editor Comments:

1 Line 139- Subsequently, - Edited as suggested

2 182 - ...who chose...- Edited as suggested

3 260 - ...third AE, also moderate, was...- Edited as suggested

4 270 Removals were... beyond). repeats lines 206-209 - could be paraphrased or left out - Paraphrased

5 326 ...locations,...??- Edited as suggested

6 366 ...uptake of ShangRing…- Edited as suggested

7 382 ....realization of public...-Edited as suggested

8 384 ...adolescents 10-12 years..- Edited as suggested

---

## [Editor Report · Decision Letter 2]

11 Sep 2019

Rollout of ShangRing circumcision with active surveillance for adverse events and monitoring for uptake in Kenya

PONE-D-19-15563R2

Dear Dr. Odoyo-June,

We are pleased to inform you that your manuscript has been judged scientifically suitable for publication and will be formally accepted for publication once it complies with all outstanding technical requirements.

With kind regards,

Webster Mavhu

Academic Editor

PLOS ONE
---

## [Editor Report · Acceptance letter]

19 Sep 2019

PONE-D-19-15563R2 

Rollout of ShangRing circumcision with active surveillance for adverse events and monitoring for uptake in Kenya 

Dear Dr. Odoyo-June:

I am pleased to inform you that your manuscript has been deemed suitable for publication in PLOS ONE. Congratulations! Your manuscript is now with our production department. 

With kind regards,

on behalf of

Dr. Webster Mavhu 

Academic Editor

PLOS ONE